# Evaluation of Weather Yield Index Insurance Exposed to Deluge Risk: The Case of Sugarcane in Thailand

Thitipong Kanchai [1], Wuttichai Srisodaphol [2], Tippatai Pongsart [2,*] and Watcharin Klongdee [1]

1 Department of Mathematics, Khon Kaen University, Khon Kaen 40002, Thailand; thitipong_p@kkumail.com (T.K.); kwatch@kku.ac.th (W.K.)
2 Department of Statistics, Khon Kaen University, Khon Kaen 40002, Thailand; wuttsr@kku.ac.th
* Correspondence: tipppo@kku.ac.th

**Abstract:** Insurance serves as a mechanism to effectively manage and transfer revenue-related risks. We conducted a study to explore the potential financial advantages of index insurance, which protects agricultural producers, specifically sugarcane, against excessive rainfall. Creation of the index involved utilizing generalized additive regression models, allowing for consideration of non-linear effects and handling complex data by adjusting the complexity of the model through the addition or reduction of terms. Moreover, quantile generalized additive regression was deliberated to evaluate relationships with lower quantiles, such as low-yield events. To quantify the financial benefits for farmers, should they opt for excessive rainfall index insurance, we employed efficiency analysis based on metrics such as conditional tail expectation (CTE), certainty equivalence of revenue (CER), and mean root square loss (MRSL). The results of the regression model demonstrate its accuracy in predicting sugar cane yields, with a split testing $R^2$ of 0.691. MRSL should be taken into consideration initially, as it is a farmer's revenue assessment that distinguishes between those with and those without insurance. As a result, the GAM model indicates the least fluctuation in farmer income at the 90th percentile. Additionally, our study suggests that this type of insurance could apply to sugarcane farmers and other crop producers in regions where extreme rainfall threatens the financial sustainability of agricultural production.

**Keywords:** crop insurance; weather derivatives; sugarcane yield; rainfall index insurance





## 1. Introduction

One of the major problems for farmers is their agricultural products being ruined due to natural factors. This issue can be caused by weather conditions, which cannot be controlled. As a result, the government must play a role to help and relieve farmers who face difficulties. Nonetheless, this does not help thoroughly and is not sufficient to attenuate the effects of the damage because the government has a limited budget and complex procedures. Therefore, help and relief for farmers may be delayed. Although farmers currently have some risk management tools, such as setting up savings cooperatives or groups to help each other, these tools are not sufficiently effective to solve the problem. Crop insurance is an alternative method that plays an important role in managing and transferring farmers' financial risks. It also helps the government to solve the problem of the high costs of helping farmers.

At present, there are many variants of crop insurances according to the insured purposes and nature of risk, such as area–yield insurance, satellite imagery in agricultural insurance, and weather index insurance (Vroege et al. 2019). Crop insurance protects insured farmers against damage or loss under the terms of the policy that the insured individual has purchased. One problem that is on the rise and is a concern for farmers worldwide is the weather problem. Climate variability is a major cause of crop loss which affects farmers' financial conditions. This has resulted in the development of insurance

products that cover crop damage from weather conditions. Consequently, crop insurance using a weather index is a type of insurance that has been researched and tested for the benefit of farmers and insurance companies. There are many advantages of weather index insurance, for example, the operating costs are relatively low and it is uncomplicated and transparent compared to other types of crop insurance (Conradt et al. 2015; Kath et al. 2018, 2019; Martin et al. 2001; Vedenov and Barnett 2004; Wang et al. 2022). In 2007, Food and Agriculture reported that Thailand started employing crop insurance, utilizing weather index insurance (Hnin Ei Win 2016). The most common weather index used as a measure of compensation for insured individuals is the rainfall index, also known as drought index insurance, because drought and flooding are the main issues for Thai agriculturists. In general, farmers who purchase crop insurance using a rainfall index will receive compensation if the percentage of rain or drought index satisfies the policy requirements.

Agriculture in Thailand is a very competitive and diverse subsector, and its exports are highly successful internationally. Various crops affect Thailand's economy, such as rice, sugarcane, and cassava, which account for more than a half of Thailand's cultivated area. In 2022, Thailand was the world's fourth largest sugarcane-producing country, with production having increased by 8.69% from the previous year (United States Department of Agriculture 2022). Moreover, in 2022, Thailand was ranked the second largest sugar exporter in the world, following Brazil (Office of Agricultural Economics 2022). Sugarcane was selected as the sample crop for this study because it is crucial to Thailand's economy, and it is sensitive to weather conditions.

In terms of sugarcane production, although Thailand is located in an area with a suitable climate for growing sugarcane, in recent years sugarcane production has faced issues, with low yields per rai (1 rai = 1600 square meters) due to weather changes and natural disasters which lower production efficiency. Rainfall is an important variable for sugarcane production because sugarcane yields vary with precipitation levels in different periods of the growing season (Pipitpukdee et al. 2020; Sinnarong et al. 2019, 2022). This means that crop insurance using a rainfall or drought index plays a role and could greatly reduce the risks faced by sugarcane farmers. Therefore, developing a crop insurance system using the rainfall index or the dryness index is important in order to assist and diversify farmers' risks. This leads to the tasks comprising this study. The first task is to determine how to formulate the crop yield model using a weather index, such as rainfall index, alongside other climate factors. The second task is to establish an insurance contract by considering the cost-effectiveness of crop insurance.

To complete the first task, sugarcane production, including the weather index, is used to create statistical models of crop yield. Temperature and rainfall are important factors contributing to climate variability in several studies (Carvalho et al. 2015; Greenland 2005; Singh et al. 2021; Verón et al. 2015; Mali et al. 2014; Verma et al. 2019, 2021). However, establishing a connection between productivity and climate is a challenging task. The essential conditions are the types of crops and the location of study. The study area in this research is Khon Kaen province, which is a significant producer of sugarcane (Office of the Cane and Sugar Board 2022). Moreover, most sugarcane growing areas are outside the irrigated region. To develop strategies to calculate suitable insurance premiums for farmers, first we utilize statistical models, including the generalized additive model (GAM) and quantile generalized additive model (QGAM), in order to obtain appropriate weather–yield models.

GAMs can be understood as Generalized Linear Models (GLMs) that are estimated while being subjected to smoothing penalties. A GAM enables the consideration of non-linear effects and the management of complex data by adjusting model complexity, changing its inside conditions. Additionally, QGAM was developed by Fasiolo et al. (2020) and can be utilized to assess relationships with lower quantiles, like occurrences of low yield. Before constructing the weather–yield model, we consider the effects that time series might have on yield productivity. Detrending the yield is one of the most popular methods for removing the effects of changes in management, technology, and production extent

on yields over time. One of the techniques for detrending the yield is adjusting only the yield variable in the model (Vedenov and Barnett 2004; Johari et al. 2022). An alternative approach can be used, because the annual impacts do not need to be identical, in which the yield data are implicitly adjusted for trends by incorporating the year as part of the model (Kath et al. 2018, 2019; Verón et al. 2015; Verma et al. 2021). The meteorological factors generally show upward temporal trends. The insurance contracts corresponding to the suitable weather–yield models are then generated to serve the second task of the study. The contracts are accessible for use by insurance companies and give adequate compensation corresponding to the individual farmer's risk.

The design of weather index insurance and the consideration behind our choice of analysis are detailed in the next section. The subsequent section presents the results of weather–yield modeling. An efficiency analysis of the models is also described in Section 3. Section 4 discusses the efficiency of the model and the implications of the study for climate risk management. Conclusions are then drawn and presented in the final section.

## 2. Materials and Methods

### 2.1. Study Area and Panel Dataset

In this study, Khon Kaen province was selected as a sugarcane cultivation area located in northeastern Thailand, as outlined in Figure 1. This figure displays reported sugarcane acreage and yield statistics over the past 31 years (1992–2022). This region is one of the most significant places in Thailand for the production of sugarcane, which covers a large area with 651,492 rai of planted sugarcane (Office of the Cane and Sugar Board 2022). Moreover, this region has a relatively high level of climate change. The weather in Khon Kaen is classified as a tropical savanna, with dry and extremely warm winters. April is an extremely hot month, with an average daily maximum temperature of 36.3 °C (97.3 °F). The monsoon season, which lasts from May to October, is characterized by heavy rains. Rainfall in the rainy season is especially heavy, causing frequent flooding.

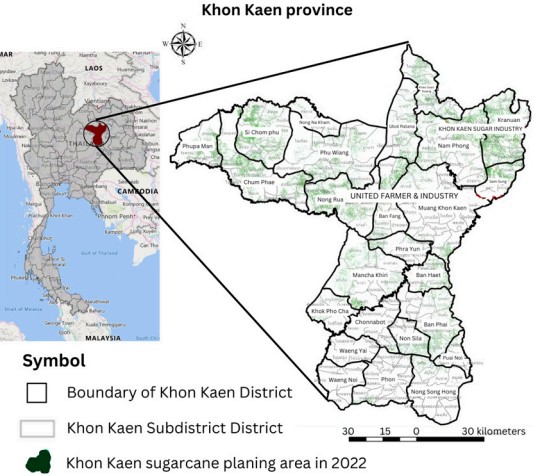

**Figure 1.** The Khon Kaen sugarcane region. In the map of Khon Kaen, the green area represents suggested areas for sugarcane cultivation in 2022.

The Office of Agricultural Economics provides annual data on sugarcane yield and farm gate prices at the provincial level from 1992 to 2022 (The Office of Agricultural Economics). Monthly climate data were obtained from the Khon Kaen meteorological station (381,201) from January 1992 to December 2022 (Thai Meteorological Department 2022).

### 2.2. Conceptual Framework

There are many studies of agricultural production or yield modeling that use the weather condition index in combination with various other factors. The reason for creating numerous models is that productions change between different areas depending on various

factors. A suitable model in one area may not be suitable in others. Firstly, the yield models using the weather index are formulated with various statistical tools according to the selected cultivation area. Subsequently, the appropriate models are used for offering crop insurance. The effectiveness of crop insurance model is directly linked to the weather–yield model efficiency. If the model has high accuracy, the crop insurance model is also of high quality and advantageous. This study offers designs and prices for the weather derivative in order to obtain suitable premiums, corresponding to the yield model. Consequently, the efficiency of the model is measured in scenarios in which farmers are either insured or not insured. Figure 2 outlines the research methodology.

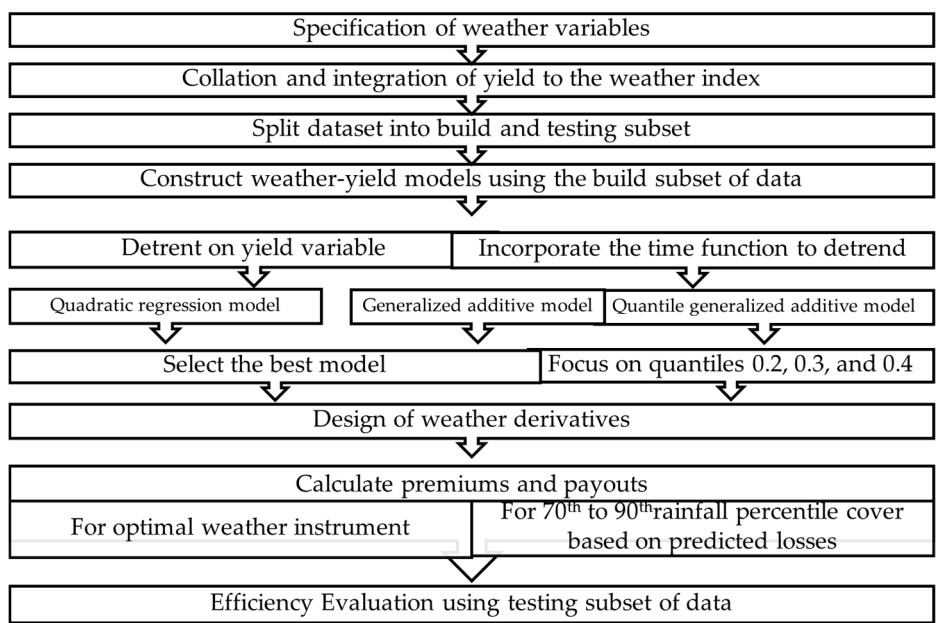

**Figure 2.** Schematic of weather index insurance design process and efficiency evaluation.

### 2.2.1. Step 1: Specification of Weather Variables

A review of the literature on the weather index related to crop yield is presented in Table 1. Many studies show that the weather index has a significant influence on crop yields. According to Table 1, rainfall and maximum temperature variables are most frequently used in the specification of weather variables for sugarcane. These two variables were therefore selected to create weather–yield models in the study. The model of the relationship between climate variables and crop yields is obtained using a statistical method to estimate the relationship between climate and crop production.

**Table 1.** Summary of the literature on climate change variability for crop yield.

| References | RI | T | T.MIN | T.MAX | SR | Other Features |
|---|---|---|---|---|---|---|
| (Martin et al. 2001) | X | X | | | | $CO_2$ |
| (Vedenov and Barnett 2004) | X | X | | | | |
| (Lobell and Field 2007) | X | X | X | X | | |
| (Lobell and Burke 2010) | X | X | | | | |
| (Verón et al. 2015) | X | X | X | X | | |
| (Wang et al. 2018) | | X | | | | SH |
| (Xu et al. 2018) | X | X | | | | SH |
| (Shirsath et al. 2019) | X | | | | | |
| (Sinnarong et al. 2019) | X | X | | | | |
| (Amnuaylojaroen et al. 2021) | X | X | | | X | |
| (Kath et al. 2021) | X | X | | | | |
| (Bucheli et al. 2022) | | X | | | | |

**Table 1.** *Cont.*

| References | RI | T | T.MIN | T.MAX | SR | Other Features |
|---|---|---|---|---|---|---|
| (Tappi et al. 2023) | X | | X | X | | DTR |
| (Greenland 2005) * | X | | X | X | | GDDs, FH, Soil water |
| (Mali et al. 2014) * | X | X | X | X | | RH.Max, RH.Min |
| (Sattar et al. 2014) * | X | X | X | X | X | |
| (Carvalho et al. 2015) * | X | X | X | X | | |
| (Kath et al. 2018) * | X | | | | | |
| (Verma et al. 2019) * | X | | X | X | | RH I, RH II |
| (Pignède et al. 2021) * | X | X | X | X | | NDVI, PE, MRH |
| (Singh et al. 2021) * | X | | X | X | X | CDC |
| (Pipitpukdee et al. 2020) * | X | X | | X | | Max.rain, PD, LRP, LW, IA |
| (Verma et al. 2021) * | X | | X | X | | RH |
| (Sinnarong et al. 2022) * | X | X | | | | |

Note: The notation (*) after references represents works involving sugarcane. RI: annual rainfall (mm); T: temperature; DTR: average diurnal temperature range; T.MIN: minimum temperature (°C); T.MAX: maximum temperature (°C); SH: sunshine hours (hour); MRH: mean relative humidity (%); PE: potential evapotranspiration; RH I: relative humidity at 8:30 IST; RH II: relative humidity at 14:30 IST; GDDs: growing degree days (GDDs) (°C day); FH: fall (autumn) hurricanes and tropical storms; SR: solar radiation (%); CDC: carbon dioxide concentration (ppm); PD: population density; LRP: lag received price (USD/ton); LW: lag wage (USD); IA: % irrigated area per province area.

### 2.2.2. Step 2: Integration of Data and Data Preparation

Climate data in the growing season, i.e., maximum temperature and rainfall, were adopted to develop the weather–yield model. Price data are taken from farm gate prices from 1992 to 2022. Details on the variables and calculation methods are in Table 2.

**Table 2.** Description of the data used in the study.

| Variable | Notation | Details |
|---|---|---|
| Sugarcane yield (tonne/rai) | $Y_t$ | Sugarcane yield at year $t$ |
| Rainfall (mm) | $RI_t$ | Cumulative rainfall in the growing season (between October of year $t-1$ and November of year $t$) |
| Maximum temperature (°C) | $Tmax_t$ | Average maximum temperature in the growing season (between October of year $t-1$ and November of year $t$) |
| Year of harvest | $t$ | $t$ stands for the year of harvest |
| Price of sugarcane yield per tonne | $P$ | Farm gate price of the last harvest year |

In some cases there are missing data, which could have resulted from various factors, such as a power cut. Data preprocessing was used to deal with this issue. From our historical data, only the rainfall index has missing data. As a result, we employed the work of Kanchai et al. (2023) to handle this problem.

### 2.2.3. Steps 3–5: Regression Model Linking Yield to Climate Indices

This study was conducted using data from 1992 to 2022. Data from 1992 to 2017 were used to fit the models, while the remaining data (2018–2022) were used for model validation. Most analytical techniques are carried out with "R version 4.0.5" and Mathematica 11.2 software.

In our study, the weather–yield model was formulated using different statistic tools, including a quadratic regression model, Generalized Additive Models (GAMs), and Quantile

Generalized Additive Model (QGAM). The quadratic regression models are widely used to create yield models, especially where adjusted yield is treated as a variable (Johari et al. 2022). Since several factors influence agricultural yields, removing non-weather influences from agricultural yield data throughout the course of the observation period is the goal of detrending crop output. For example, the factors include geography, governance policies, and technological development.

Alternative approaches in the study are to use GAMs or a QGAM. These models were utilized by inserting the function of year into the model. The primary challenge that arises from this penalization is the requirement to choose the appropriate level of penalization, meaning that the smoothing parameters must be estimated. To analyze and predict productivity in the study, GAMs are utilized corresponding to the following equation:

$$Y_t = a + s(RI_t) + s(Tmax_t) + s(t) + \varepsilon_t, \tag{1}$$

where $Y_t$ is sugarcane yield at time $t$, $s$ is smooth function, $RI_t$ is the rainfall index at time $t$, $\varepsilon_t$ is the error term, and $t$ stands for year.

Instead of focusing only on the mean, as in the case with the GAM, the rainfall index relationship with the lower quantiles of sugar cane yield distribution is evaluated using the quantile generalized additive model. The equation of QGAM considered in this study is as same as the GAM as shown in (1).

It is possible that the model could be overfitted in the model construction step. It is possible to check whether the model is overfitted. For this purpose, the data are divided into a training and a testing subset. After that, the split test technique is employed to check the overfitting. Next, we measure the quality of the fitting model. The adjusted $R^2$ value was used as a criterion (greater than 0.6 in the study) for model selection.

### 2.2.4. Steps 6 and 7: Design of Weather Derivatives and Premium Estimation

To evaluate the efficiency of weather derivatives, a particular contract must be designed. From the literature in Table 1, the rainfall index is desired as the main factor affecting sugarcane production, especially in excessive rainfall events (Glaz and Lingle 2012; Gomathi et al. 2015). Thus, in this study, an elementary contract in case of excessive rainfall triggered by rainfall index is proposed (adapted from Vedenov and Barnett 2004). This contract pays an optimal indemnity conditional on the realization of an index, as depicted in Figure 3.

$$\text{f}\left(i\middle| x, i^{(\alpha)}, L\right) = x \times \begin{cases} 1 & ; i > Li^{(\alpha)}, \\ \frac{\left(i - i^{(\alpha)}\right)}{Li^{(\alpha)} - i^{(\alpha)}} & ; i^{(\alpha)} < i < Li^{(\alpha)}, \\ 0 & ; i \leq i^{(\alpha)}, \end{cases} \tag{2}$$

where $i^{(\alpha)}$ is the rainfall index with $\alpha$ percentile levels and $x$ denotes maximum indemnity. $L$ is a constant, where $Li^{(\alpha)}$ is the limit of rainfall at which the maximum insurance claim is paid.

We assume that potential insurance customers will not take any risks at all, so they will be compensated if there is a loss in sugarcane yield the following year. From Figure 3, the contract starts to pay whenever the index $i$ raises above a specific rainfall index with $\alpha$ percentile levels $i^{(\alpha)}$. The maximum indemnity (equal to $x$) is paid whenever the index raises above the limit $Li^{(\alpha)}$, where $0 < L < 1$. Therefore, an elementary contract can be identified by fixing three parameters: the rainfall index with $\alpha$ percentile levels, limit, and maximum indemnity.

To price an elementary contract, the contract parameters as well as the probability distribution of the essential index must be determined. For weather derivatives, the distribution can be derived based on historical data by using either the parametric or

non-parametric approach. If $h(i)$ is a probability density function of the index, the expected payoff (equal to pure premium) of the contract can be determined by:

$$\pi(x, L) = x \int_{Li^{(\alpha)}}^{\infty} h(i)di + x \int_{i^{(\alpha)}}^{Li^{(\alpha)}} h(i) \frac{\left(i - i^{(\alpha)}\right)}{Li^{(\alpha)} - i^{(\alpha)}} di \tag{3}$$

notice that for any of the rainfall indices with $\alpha$ percentile levels $i^{(\alpha)}$ and limit parameter $Li^{(\alpha)}$, if $x$ is the price of a contract with maximum indemnity, then $\pi(x, L)$ is the premium of a contract with maximum indemnity of $x$ baht.

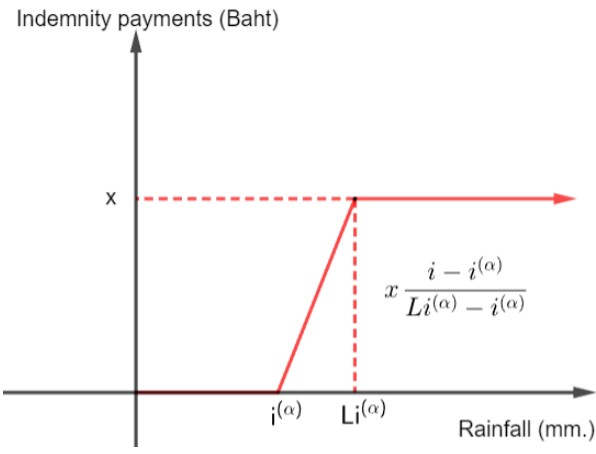

**Figure 3.** Payoff design of an element contract.

Selection of Contract Parameters

The weather–yield model resulting from Step 5 was used to calculate payoffs in insurance contracts. At this point, a rainfall index as a trigger value of the contract must be determined. The rainfall index at strike level $i^{(strike)}$ can be selected in the sense that the predicted yields were equal to the corresponding long-time yield averages. The remaining parameters, i.e., the limit parameter and the optimal number of B1 contracts, were selected in order to minimize an aggregate measure of downside loss. In addition, the parameters $x$ and $L$ were determined to be solved (adapted from Vedenov and Barnett 2004):

$$\min_{x, L} \sum_{t=1}^{26} \left( \max\left\{ P(\overline{Y} - Y_t) - f\left(i \middle| x, i^{(strike)}, L\right) + \pi(x, L), 0 \right\}^2 \right), \tag{4}$$

where $P$ denotes the farm gate price in the last harvest year (in the training subset, i.e., the year 2017), $Y_t$ is the yield at time $t$, $f\left(i \middle| x, i^{(strike)}, L\right)$ is the insurance payout for that strike level of insurance, and $\pi(x, L)$ is the price of a contract with a maximum indemnity of $x$ baht.

Furthermore, we are interested in a rainfall index at different percentile levels of $\alpha$, $i^{(\alpha)}$ exposed to excessive rainfall risk (i.e., $\alpha = $ 70th, 80th, and 90th percentiles of the rainfall index). The payoffs and premium of the contracts corresponding to these percentile levels are illustrated for the purpose of comparison to the strike level. The efficiencies of these contracts for excessive rainfall risk were also examined.

2.2.5. Step 8: Efficiency Analysis

Efficiency analysis methods were adapted from Kath et al. (2018). Three methods were used, including conditional tail expectation (CTE), certainty equivalence revenue (CER), and mean root square loss (MRSL), to assess the benefit of the contract. To analyze the impact of the contract on farmer revenue, we compared revenue with and without insurance at $\alpha$ level of the rainfall index $i^{(\alpha)}$. We derived the parameters of optimal contracts and measured the in-sample performance (adapted from Vedenov and Barnett 2004). In

addition, at different percentile coverage levels for the selected regression model, the revenue without a contract is given by:

$$R_t = PY_t \tag{5}$$

and with a contract is:

$$R_t^{(\alpha)} = PY_t + f\left(i_t \middle| x, i^{(\alpha)}, L\right) - \pi\left(x, i^{(\alpha)}, L\right), \tag{6}$$

where $R_t$ denotes revenue at time t without insurance, $P$ is the farm gate price, $R_t^{(\alpha)}$ is revenue at time $t$ with $\alpha$ percentile levels of insurance (here the strike level and the 70th, 80th, and 90th percentiles of the excessive rainfall index), $Y_t$ is yield at time $t$, $f\left(i_t \middle| x, i^{(\alpha)}, L\right)$ is the insurance payout for that level of insurance in that year (predicted from the regression models), and the yearly premium for $\alpha$ percentile levels of insurance is constant throughout the years in question, so it is written as $\pi\left(x, i^{(\alpha)}, L\right)$.

Conditional tail expectation (CTE) is a method to assess the effectiveness of insurance in protecting against financial risks. The certainty equivalence revenue with $\alpha$ percentile levels of insurance ($CTE^{(\alpha)}$) has the following equation:

$$CTE^{(\alpha)} = \frac{1}{T}\sum_{i=1}^{T} R_t^{(\alpha)} \tag{7}$$

next, certainty equivalence revenue (CER) was used to measure willingness to pay the farmer, assuming that the risk aversion was taken into account and the constant relative risk aversion was assumed. Therefore,

$$CER^{(\alpha)} = \frac{1}{T}\sum_{i=1}^{T} ln(R_t^{(\alpha)}), \tag{8}$$

where $CER^{(\alpha)}$ is the certainty equivalence revenue with $\alpha$ percentile level of insurance.

Finally, mean root square loss (MRSL) is a value that represents an insurance contract's ability to mitigate risk. MRSL with $\alpha$ percentile levels of insurance was calculated as,

$$MRSL^{(\alpha)} = \sqrt{\frac{1}{T}\sum_{t=1}^{T}\left[max\left(P\overline{Y} - R_t^{(\alpha)}, 0\right)\right]^2}, \tag{9}$$

where $P$ is the price of agricultural commodity and $\overline{Y}$ is the long-term average yield.

## 3. Results

### 3.1. Quadratic Regression Modeling Results

The adjusted sugarcane yield is computed based on the estimated sugarcane yield of the last year in the data range. The final year's sugarcane yield level is anticipated to be the most similar to the output of the next year, since the yield increases steadily over time. Therefore, the next year's sugarcane yield may be predicted using the modified sugarcane yield. Since 2022 is the last year for which the sugarcane yield was projected in our instance, the adjusted yield will indicate the quantity of sugarcane output anticipated for 2023 based on 2022. However, data should be gathered up until the current year in order to build a true insurance contract. At that point, the modified sugarcane yield may be useful for the current year. The crop yield data will be detrended by fitting data. A quadratic regression model was tested to detrend the time trend in crop yield. The results are shown in the following equation:

$$Y_t^{trend} = \alpha_0 + \alpha_1 t + \alpha_2 t^2, \tag{10}$$

where $Y_t^{trend}$ is sugarcane yield at $t$ and $t = 1, 2, \ldots, 31$ stands for year = 1992, 1993, $\ldots$, 2022.

This quadratic regression model explained 10.14% of the yield variation. This regression model estimates the intercept of time trend poorly, indicating that it is not a good model to fit the data for sugarcane yield. However, in order to illustrate the relationship between an adjusted sugarcane yield and weather indexes consisting of RI and T.MAX, the GAM was employed to construct a weather–yield model resulting in Equation (11):

$$Y_t^{detrend} \sim s(RI) + s(T.MAX, bs = \text{``re''}, k = 8). \tag{11}$$

From Equation (11), the model with an adjusted $R^2$ value equal to 0.284 is inadequate for use. Therefore, we should neglect this model and it is not necessary to measure the efficiency of the model in Equation (11) in the next step.

### 3.2. GAM Regression Modeling Results

GAMs can apply to non-linear models built using a spline. By letting time trend interpretation serve as a function for a model, a GAM was constructed corresponding to a trained data set. Furthermore, to improve the accuracy of the model, the internal structure of the model was restructured as shown in Equation (12):

$$Y_t \sim s(RI, bs = \text{``cc''}, k = 9) + s(T.MAX, bs = \text{``re''}, k = 8) + s(t, bs = cc, k = 9), min.sp = c(0.001, 0.01, 0, 10). \tag{12}$$

The split testing method for the regression model revealed that it explained 69.1% of the yield variation (Table 3). Figure 4 shows how the GAM model predicted yield responses to the rainfall index. Grey-shaded regions indicate 95% confidence intervals.

**Table 3.** Sugar cane model with 31 years of harvest as dependent variable results with the approximate significance of smooth terms for predictors.

| Predictor Variable | *F* | *p*-Value |
|---|---|---|
| Year | 9.319 | 0.0007 *** |
| Rainfall index | 1.625 | 0.0444 * |
| Maximum temperature | 0.284 | 0.0976 |
| Adjust $R^2$ | 0.602 | |
| Split testing $R^2$ | 0.691 | |

Significant at *** $p < 0.001$, * $p < 0.05$.

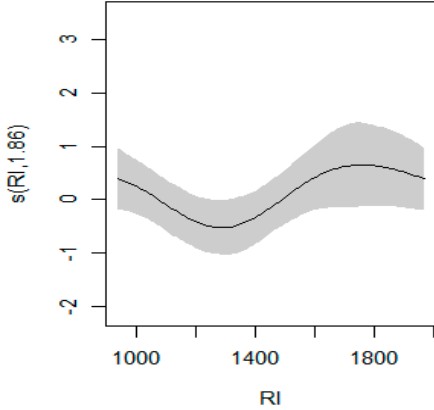

**Figure 4.** Predicted yield responses to the rainfall index from the GAM model.

### 3.3. QGAM Regression Modeling Results

QGAM for the response variable sugarcane yield at the time *t* was fitted with a smoothing function (*s*) for the rainfall index, the maximum temperature, and the year. Moreover, we adjusted the spline in the smooth function for better results as shown in the equation below.

$$Y_t \sim s(RI, bs = \text{tp}, k = 9) + s(T.MAX, bs = \text{tp}, k = 9) + s(t). \tag{13}$$

Since we are interested in low-yield event responses to the excessive rainfall index, we concentrate on quantiles 0.4, 0.3, and 0.2. For each tau level (0.2, 0.3, and 0.4) examined with the QGAM, the rainfall index was significant at $p = 0.1$ as shown in Table 4.

**Table 4.** Results of the QGAM model.

| Tau | Predictor Variable | *p*-Value |
|---|---|---|
| **0.4** | Year | <0.0001 *** |
| | Rainfall index | 0.0535 |
| | Maximum temperature | 0.0740 |
| **0.3** | Year | <0.0001 *** |
| | Rainfall index | 0.0757 |
| | Maximum temperature | 0.0657 |
| **0.2** | Year | <0.0001 *** |
| | Rainfall index | 0.0036 ** |
| | Maximum temperature | 0.0960 |

Significant at *** $p < 0.001$, ** $p < 0.01$.

### 3.4. Estimated Insurance Premiums and Revenues

First, the probability density function must be determined, which is used in Equation (3) in Section 2.2.4 to calculate the expected payoff of the contract. The rainfall data indicate that the distribution of this rainfall has a gamma distribution with $(\alpha, \beta, \gamma) = (22.435, 60.319, 0)$. Next, the parameters $x$ and $L$ were settled in Equation (4). The equation introduces the idea that the indemnity amount in the insurance contract depends on the sugarcane yield strike level, which is determined by the weather index. Therefore, we just need to set the strike level $i^{(strike)}$ of the rainfall index. Since we presume that the prospective insurance customers are completely risk averse, they will profit in the event that the sugarcane yield declines the following year. The average sugarcane yield from our data history should serve as the strike level. Then $i^{(strike)}$ was calculated. The strike levels, including the different percentiles of rainfall index for the GAM and QGAM corresponding to Equations (12) and (13), respectively, are shown in Table 5. Consequently, the weather-index-based crop insurance was designed as follows in the table below.

**Table 5.** Parameters of optimal weather instruments.

| Model | Levels of the Excessive Rainfall | Maximum Liability (baht/Rai) | Premium (baht/Rai) | Premium Rate (%) |
|---|---|---|---|---|
| GAM | 1595.42 (strike) | 1216.60 | 4.70369 | 0.38663 |
| | 1492.75 (70th) | 1204.61 | 6.44293 | 0.53486 |
| | 1573 (80th) | 1213.06 | 5.04298 | 0.41572 |
| | 1789.3 (90th) | 1239.79 | 2.50249 | 0.20185 |
| QGAM Tau = 0.4 | 1595.42 (strike) | 1704.37 | 6.58927 | 0.38661 |
| | 1492.75 (70th) | 1686.47 | 9.01907 | 0.53479 |
| | 1573 (80th) | 1700.78 | 7.07056 | 0.41573 |
| | 1789.3 (90th) | 1726.82 | 3.48553 | 0.20185 |
| QGAM Tau = 0.3 | 1595.42 (strike) | 1733.34 | 6.70155 | 0.38663 |
| | 1492.75 (70th) | 1714.23 | 9.16753 | 0.53479 |
| | 1573 (80th) | 1729.55 | 7.19016 | 0.41572 |
| | 1789.3 (90th) | 1756.32 | 3.54507 | 0.20185 |
| QGAM Tau = 0.2 | 1595.42 (strike) | 1814.64 | 7.01588 | 0.38663 |
| | 1492.75 (70th) | 1795.24 | 9.60079 | 0.53479 |
| | 1573 (80th) | 1811.14 | 7.52931 | 0.41572 |
| | 1789.3 (90th) | 1838.04 | 3.71002 | 0.20185 |

The premium rate is the ratio of premium to maximum liability.

From Table 5, the percentile cover and regression method had a significant impact on premium variations. The most costly premiums (9.60079 baht/Rai) came from the QGAM at tau = 0.2 at the 70th percentile cover level, while the least expensive premiums (2.50249 baht/Rai) were estimated from the GAM at the 90th percentile level of cover. According to Table 5, the maximum liability was calculated to be the lowest for the GAM at 1204.61 baht/Rai and the highest for the QGAM at tau = 0.2 at 1838.04 baht/Rai. Table 5 shows that premium rates varied from approximately 0.2% to 0.5%. The 90th percentile cover had the lowest premium rates for each model, while the 70th percentile cover had the highest premium rates (Table 5).

In order to illustrate the influence of an insurance contract on a farmer's revenue, examples of revenue differences for each of the different levels of rainfall in the GAM model that we carried out are shown in Figure 5. Revenues below 0 indicate a year where a premium was paid, but no payout was received. Revenues above 0 indicate a year when the rainfall index triggered a payout.

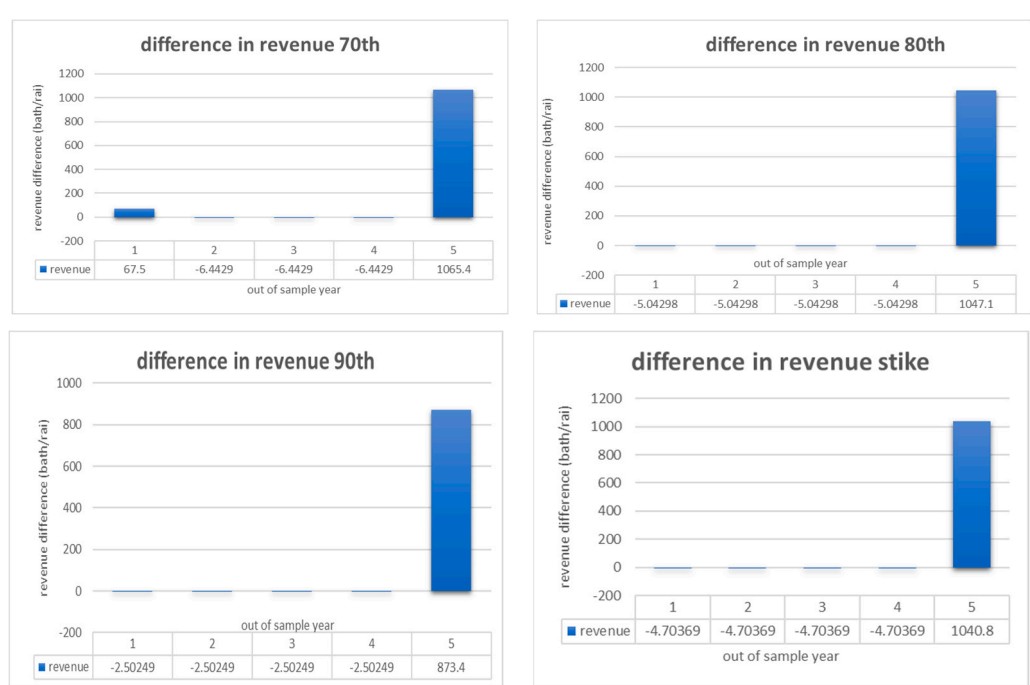

**Figure 5.** Examples of difference in revenue.

### 3.5. Efficiency Analysis of Weather Index

The efficiency of our insurance model was tested according to Equations (7)–(9). The results are shown in Table 6. Based on the CTE, CER, and MRSL efficiency analyses, the excessive rainfall index insurance for sugarcane was found to provide advantages. The most significant potential advantages for farmers were indicated by the 90th percentile cover predicted from the GAM and QGAM (0.2, 0.3, 0.4), which protects them from excessive rainfall due to MRSL results. The QGAM model with tau = 0.2 at the 70th percentile demonstrates the highest benefit, whereas the GAM model at the 90th percentile is the least beneficial due to CTE and CER results.

**Table 6.** Efficiency of rainfall index as measured by CTE, CER, and MRSL.

| GAM | In Sample (1992–2017) | | | Out of Sample (2018–2022) | | |
|---|---|---|---|---|---|---|
| | **CTE** | **CER** | **MRSL** | **CTE** | **CER** | **MRSL** |
| 70th | 6918.65 | 8.7275 | 603.900 | 10,497.44 | 9.2389 | 1410.261 |
| 80th | 6895.68 | 8.7237 | 607.180 | 10,480.12 | 9.2376 | 1409.380 |
| 90th | 6861.85 | 8.7187 | 626.675 | 10,447.41 | 9.2350 | 1407.782 |
| Strike | 6891.53 | 8.7230 | 608.646 | 10,479.13 | 9.2375 | 1409.167 |
| **QGAM (0.2)** | In Sample (1992–2017) | | | Out of Sample (2018–2022) | | |
| | **CTE** | **CER** | **MRSL** | **CTE** | **CER** | **MRSL** |
| 70th | 6992.90 | 8.7381 | 599.050 | 10,606.64 | 9.2475 | 1412.251 |
| 80th | 6958.33 | 8.7325 | 601.313 | 10,581.39 | 9.2455 | 1410.948 |
| 90th | 6905.63 | 8.7249 | 626.675 | 10,526.91 | 9.2413 | 1410.948 |
| Strike | 6951.80 | 8.7314 | 602.721 | 10,579.09 | 9.2454 | 1410.948 |
| **QGAM (0.3)** | In Sample (1992–2017) | | | Out of Sample (2018–2022) | | |
| | **CTE** | **CER** | **MRSL** | **CTE** | **CER** | **MRSL** |
| 70th | 6982.72 | 8.7367 | 599.417 | 10,591.67 | 9.2463 | 1411.978 |
| 80th | 6949.79 | 8.7313 | 601.933 | 10,567.57 | 9.2445 | 1410.734 |
| 90th | 6899.65 | 8.7241 | 626.675 | 10,515.70 | 9.2404 | 1410.734 |
| Strike | 6943.61 | 8.7303 | 603.383 | 10,565.46 | 9.2443 | 1410.734 |
| **QGAM (0.4)** | In Sample (1992–2017) | | | Out of Sample (2018–2022) | | |
| | **CTE** | **CER** | **MRSL** | **CTE** | **CER** | **MRSL** |
| 70th | 6979.23 | 8.736 | 599.564 | 10,586.53 | 9.2460 | 1411.885 |
| 80th | 6946.77 | 8.731 | 602.165 | 10,562.70 | 9.2441 | 1410.659 |
| 90th | 6897.49 | 8.724 | 626.675 | 10,511.66 | 9.2401 | 1410.659 |
| Strike | 6940.69 | 8.730 | 603.630 | 10,560.60 | 9.2439 | 1410.659 |

## 4. Discussion

### 4.1. Crop Yield Models and Efficiency of Weather Index Insurance for Sugarcane

We employed regression models to evaluate sugarcane yields using weather condition indices in order to create our crop insurance model. These require determining the right insurance price and assessing the crop insurance contract's effectiveness. Data from any cultivation area could be used to build the model. As a result, the regression model may be adjusted based on the data. From the data received, the data of rainfall and sugarcane production are complicated and disorganized. Before generating the weather–yield model, the impact of time series on yield productivity was taken into consideration. Detrending is a technique used to remove trends or non-weather-related components from a variable. In the case of the detrended yield variable, we can evaluate the adjusted R-squared to provide the effectiveness of detrending. In a case where the time function is added, the effectiveness can be measured by the adjusted R-squared of the overall model and the significance of a time variable.

Although both linear and quadratic regressions are easy to apply and show an obvious trend, the GAM and QGAM were used for this study because of their flexibility and ease of fitting curves to the data. However, these models have some limitations. For instance, the model does not meet certain criteria in practice, such as an adjusted R-squared of less than 0.6 or a variable link inconsistent with the study's objectives. Moreover, the models constructed by the GAM and QGAM cannot illustrate the exact form of equations, while the linear or quadratic regression models show equations. Alternatively, we may look at techniques like satellite imaging in agricultural insurance, area–yield insurance, spatial autocorrelation, or spatial regression. In this work, we use two weather variables, i.e., rainfall and maximum temperature, to get a more accurate crop yield model because these weather conditions affect sugarcane. Kath et al. (2018) used only rainfall as a weather variable. However, data have changed over time. It is necessary to revise the model

according to the updated data to obtain an accurate production model appropriate for further premium calculations.

Regarding weather index insurance efficiency, the assessed performance of each insurance model reveals that measuring efficiency using CTE and CER at rainfall percentiles of 70, 80, and 90, as well as at the strike level, yields slightly different results. This is partially because there is little information available on crops and weather. CTE and CER assess whether insurance will increase farmers' revenue during excessive rainfall. However, the risk reduction for farmers in terms of revenue fluctuations or stability may be determined from MRSL. Since MRSL is a farmer's revenue assessment between those with and those without insurance, MRSL should be considered first. Consequently, the GAM model is at the 90th percentile, indicating the least income variability for farmers. This differs from the findings of Kath et al. (2018), who studied sugarcane insurance under the title 'Index insurance benefits agricultural producers exposed to excessive rainfall risk'. Their research showed that measuring efficiency using CTE and CER is most effective at the 95th percentile, the highest percentile. However, the MRSL method performs best at the 70th percentile. These measurements are crucial for evaluating risk and financial impacts in situations characterized by uncertainty and volatility in agricultural data or possible financial events for farmers. In terms of crop insurance in Thailand, a recent work by Sinnarong et al. (2022) proposed an insurance policy for various crops based on the estimation of the impact of climate variables on economic crop production, for instance, rice, oil palm, and rubber. However, a sugarcane insurance was not provided. Therefore, our work can be a significant tool in managing sugarcane farmers' financial risks.

*4.2. Implications*

At present, natural disasters and excessive drought are the main focus of climate index crop insurance in several studies. Nevertheless, other climate factors could seriously affect farmers' yields for some crops. Therefore, we raised awareness of the problem of excessive rainfall during the growing season, especially for sugarcane. Additionally, this insurance is becoming more important due to increased climate change. It may become an increasingly popular and significant tool for risk management if the government seeks to promote sugarcane insurance, similar to the Thai rice insurance project in which the government helps farmers with insurance costs. The Thai rice insurance project was achieved due to cooperation between the government and the Thai General Insurance Association. Between 2011 to 2022, more than 16,492 million baht of insurance premiums were paid to insurance companies and the total claims accounted for approximately 78% of the premium (Thai General Insurance Association 2022). This emphasized the role of insurance in managing risk within agricultural economics. For the government to subsidize the cost of farmers' sugarcane insurance premiums, strike levels may be used to gauge rainfall in order to manage farmer compensation or expand their options. Premiums should then be paid based on the QGAM model, which allows for various integrations with coverage increasing or decreasing as required.

**5. Conclusions**

This study examined the effects of climate change on economic crop production using different regression models and by constructing an insurance model to protect farmer's revenue from climate change. Determining an association between the weather index and the crop, as well as discovering the point of departure and the level of the weather index that contributes to crop loss, were necessary in order to calculate the pure premium. From the modeling, a GAM and QGAM are used to model the yield of sugarcane insurance using the excessive rainfall index for the selected area. The results of this study demonstrate that sugarcane growers' financial risk decreased with crop insurance. Moreover, the yield model may improve with sufficient weather and yield data. The yield model uses a GAM and QGAM, non-linear regressions that can be used for plants sensitive to weather conditions. This yield model concept can therefore be applied to other climate-sensitive crops. In

addition, the calculation of premiums for farmers' crop insurance performance indicators, such as CTE, CER, and MRSL, are used to decide the appropriate form of crop insurance for farmers in each area. This insurance makes it easy to pay indemnity because indemnity is paid based on the amount of rain, according to the criteria specified in the insurance contract. As a result, the indemnity does not depend on the individual's actual losses. Moreover, it also reduces the cost of disaster surveys.

**Author Contributions:** Conceptualization, T.K. and T.P.; methodology, T.K. and T.P.; software, T.K.; validation, T.K., T.P. and W.K.; formal analysis, T.K.; investigation, T.K., W.S., T.P. and W.K.; resources, T.K.; data curation, T.K.; writing—original draft preparation, T.K. and T.P.; writing—review and editing, W.S., T.P. and W.K.; visualization, T.K.; supervision, T.P. and W.K.; project administration, T.P.; funding acquisition, T.P. All authors have read and agreed to the published version of the manuscript.

**Funding:** This research was funded by the Fundamental Fund of Khon Kaen University, fiscal year 2022, the National Science, Research and Innovation Fund (NSRF), Thailand.

**Data Availability Statement:** Publicly available data can be found on the provided website. https://github.com/ThitipongKan/Data-Availability/blob/main/Data%20Availability.rar, accessed on 15 July 2022.

**Acknowledgments:** The authors would like to express our heartfelt appreciation to Wikanda Phaphan for helpful advice on completing this paper successfully.

**Conflicts of Interest:** The authors declare no conflicts of interest.

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
