# Peer review of "Evaluation of Weather Yield Index Insurance Exposed to Deluge Risk: The Case of Sugarcane in Thailand"

_jrfm, doi:10.3390/jrfm17030107_

Round 1
Reviewer 1 Report
Comments and Suggestions for Authors
This type of work is very interesting, especially in the developing world where farmers desperately need protection against climate and weather-related challenges. The manuscript is interesting but the presentation needs to improve in a big way before the manuscript can be considered acceptable.
Abstract:
Whilst a reasonable background and the goal are provided, the gap in knowledge in this area of research is missing. The methods are well-summarized, but providing a short sentence supporting their choice would improve the quality of the presentation.
Introduction:
The knowledge gap is presented here. But, because there are many variants of the insurance products in the literature, it is not very clear what this one will contribute, which is not already available in the literature. The flow of ideas is not good in the sections of the introduction. [Presenting one idea per paragraph would assist]
You have disjointed facts. Expand on information covering climate variability and then expand and deal with information on crop insurance in separate paragraphs.
This section from lines 47 -51 is not clear. I propose that you rewrite those sentences. Then, move the sentence starting with "Additionally, when ...." to the be the last one in this paragraph.
Thailand has several crop insurance indices. Justify why developing this one. Indicate what are the shortfalls of the existing indices.
In this section, you need to justify or motivate the selection of QGAM, GAM, and GML for these purposes.
Material and methods:
The first sentence is repetition. lines 95-96. Please, rephrase it. Provide climate and weather facts/details for your study area. These are missing.
Include a scale bar with Figure 1.
Rewrite sentences from lines 116-120. You have excess words. It is difficult to understand that section.
Somewhere in the introduction text, specify exactly, which weather variables were used and explain why each one was selected (i.e. provide details of how each influences yields).
Justify the selection of maximum temperature and rainfall.
Proved details on how model selection was done. '[How did you get to judge the models?
Comments on the Quality of English Language
You have several minor grammatical mistakes in the manuscript. You must clean these up.
For example, Lines 64-66: Rainfall is an important variable for sugarcane production because sugarcane yields vary with precipitation levels in different periods of crop growing (Pipitpukdee et al. 2020; Sinnarong et al. 2019, 2022). ............. The word season is missing [Crop growing what?......]
Line 159: In some casts The word should read "in some cases"
Section 2.2.3
This section is overly generalizing. Be specific on what technique was used for what purpose. Briefly provide adequate summaries of the steps taken to achieve your results [GAM, QGAM, GLM]. Rewrite the section on modeling (methods]. You have vague explanations of what was done in some stages. In some cases, it is not clear why some stages were introduced or incorporated.
Results:
The presentation of the results section is not adequate. The presentation should be improved vai taking out most of the equations and mathematical
details, which can be moved to the method section. Here, you present the model performances adequately. Furthermore, provide sufficient text/details for non-mathematical readers to understand the results, especially the efficiency analysis.
Discussion:
The discussion is technically flawed and thus difficult to follow. As is it is not comprehensive.
I propose as guidance:
First, presenting the research question/objectives. Then, summarize the main findings per object and then discuss them one at a time.
Author Response
Please see the attachment
we have revised our manuscript and it has been sent to the editor already.

Reviewer 2 Report
Comments and Suggestions for Authors
Although the article is valuable, I have a few comments. I think making the following corrections will raise the scientific value of the article.
Comments:
- Due to the cross-border nature of the phenomena analysed, it is worth explaining to the readers why the use of spatial methods and the inclusion of, for example, spatial autocorrelation analysis or the use of spatial regression modelling was abandoned. Of course, I do not require the use of these tools in this work, but it may be worth considering the inclusion of these aspects in the future. Nevertheless, I propose to address the topic of possible spatial correlations.
- in the Discussion section, I propose to refer to the results of other authors obtained in similar analyses,
- in the Discussion/Conclusions section, it is worth pointing out the limitations/weaknesses of the analyses carried out
- the Conclusions section should be slightly more elaborated.
Minor ones:
- Figure 1 (Khon Kaen province) is not very clear. Its title below should be different from its current form: The Khon Kaen sugarcane region. In the map of Khon Kaen, the green area represents suggested areas for sugarcane cultivation in 2022. The red markers show sugarcane factories (???).
- renumber the figures - figure 1 appears twice
Author Response

(The authors gave the same response as above.)

Reviewer 3 Report
Comments and Suggestions for Authors
The paper presents a study on an insurance system to protect sugarcane farmers from revenue loss due to crop failure. The topic fits the profile of the journal.
The introduction is comprehensive and explains the context of the study.
The methodology presents the study area, the data that were used in the study and the mathematical models that were applied. The research methodology could be improved by carrying out correlation tests before applying the models.
The results section is clearly presented, but in my opinion it shows that the models failed to reflect the correlation between the weather parameters and yields. So I don't understand why you decided to carry on with these models instead of trying different approaches to process the data.
There should be a discussion about the limitations of the models and alternative approaches from similar studies.
The conclusions are brief and they mention that better data would be needed, but, in my opinion different analysis tools should be also considered.
Comments on the Quality of English LanguageMinor spelling mistakes
Author Response

(The authors gave the same response as above.)

Round 2
Reviewer 1 Report
Comments and Suggestions for Authors
Huge improvements are noted in the manuscript. A few issues remain to be clarified,
1) Your concluding statement in the abstract is misleading a bit. You only concluded around the ability of weather-yield models to predict yields. You need another 1-2 sentences concluding on the performance of the insurance index developed.
2) Provide additional sentences explaining the link between the insurance contracts and the yield models investigated.
3) I am not 100% convinced that the process of detrending removes all influences of non-weather variables. An R2 value of 0.69 suggests there remains unaccounted-for or unexplained variation. I expect to see this matter discussed in the discussion section.
4) In fact, provide more details to clarify what happened during the process of detrending.
5) Missing from the discussion section is a bit more of giving context to your findings. Discuss your findings in light of what others working in similar research work found. Do your results support or contradict what has been found elsewhere?
Comments on the Quality of English Language
Line 52 is vague. Please, rephrase it.
Line 131 has a grammar issue. The word should be "depending"
Reviewer 3 Report
Comments and Suggestions for Authors
The manuscript was improved
Author Response
The manuscript has been improved. We have revised and added the sentences by highlighting them in the manuscript.